# Non-Destructive Measurements for 3D Modeling and Monitoring of Large Buildings Using Terrestrial Laser Scanning and Unmanned Aerial Systems

**DOI:** 10.3390/s23125678

**Published:** 2023-06-17

**Authors:** Mircea-Emil Nap, Silvia Chiorean, Calimanut-Ionut Cira, Miguel-Ángel Manso-Callejo, Vlad Păunescu, Elemer-Emanuel Șuba, Tudor Sălăgean

**Affiliations:** 1Department of Land Measurements and Exact Sciences, Faculty of Forestry and Cadastre, University of Agricultural Sciences and Veterinary Medicine Cluj-Napoca, 400372 Cluj-Napoca, Romania; mircea.nap@usamvcluj.ro (M.-E.N.); silvia.chiorean@usamvcluj.ro (S.C.); tudor.salagean@usamvcluj.ro (T.S.); 2Departamento de Ingeniería Topográfica y Cartografía, E.T.S.I. en Topografía, Geodesia y Cartografía, Universidad Politécnica de Madrid, 28031 Madrid, Spain; ionut.cira@upm.es (C.-I.C.); m.manso@upm.es (M.-Á.M.-C.); 3Department of Mathematics, Physics and Measurements, Faculty of Land Reclamation and Environmental Engineering, University of Agronomic Sciences and Veterinary Medicine of Bucharest, 011464 Bucharest, Romania; vlad.paunescu@topoexim.ro

**Keywords:** 3D modeling, 3D monitoring, TLS, UAS

## Abstract

Along with the development and improvement of measuring technologies and techniques in recent times, new methods have appeared to model and monitor the behavior of land and constructions over time. The main purpose of this research was to develop a new methodology to model and monitor large buildings in a non-invasive way. The methods proposed in this research are non-destructive and can be used to monitor the behavior of buildings over time. A method of comparing point clouds obtained using terrestrial laser scanning combined with aerial photogrammetric methods was used in this study. The advantages and disadvantages of using non-destructive measurement techniques over the classic methods were also analyzed. With a building located in the University of Agricultural Sciences and Veterinary Medicine Cluj-Napoca campus as a case study and with the help of the proposed methods, the deformations over time of the facades of that building were determined. As one of the main conclusions of this case study, it can be stated that the proposed methods are adequate to model and monitor the behavior of constructions over time, ensuring a satisfactory degree of precision and accuracy. The methodology can be successfully applied to other similar projects.

## 1. Introduction

In the vast field of geodesy, with the increasingly rapid evolution of technology, more and more options regarding measurements have appeared. Specialists and researchers in the field have had to find more and more ingenious solutions to the problems that have arisen.

The present work was also developed in this context. This research tries to come up with a new approach regarding the use of current techniques and technologies in order to model large constructions and, especially, to monitor them, without physically intervening in their structures or even on their exterior appearance.

Regarding already established technologies, Global Navigation Satellite System (GNSS) technology makes its presence felt not only as an independent measurement technique in geodesy but is also integrated into the construction principles of other geodetic technologies [1,2]. Depending on the receivers and methods chosen, the accuracies achieved by these systems make them particularly useful for the execution of topographic surveys when they are used for the realization of support networks [3,4].

When combined or even embedded in tools such as robotic total stations, resulting in “smart station” equipment, or in tools such as Unmanned Aerial Systems (UASs) [2,5], this technology truly contributes to the development and evolution of far-field measurement works [1,4].

Modern and alternative survey solutions, such as Unmanned Aerial Systems (UAS), may be preferable to conventional surveys, depending on the purpose and desired outcome, because of their capacity to produce high-definition photographs, conduct measurements, and transmit and store data [6]. This method offers a surface representation that would be unattainable with conventional instruments because of the large number of points acquired, its adaptability, and its efficiency. In conjunction with specialized software for building accurate and complex 3D models, UAS-obtained data consist of a vast number of point clouds that provide impressive outputs and outcomes [7,8].

Drone solutions have quickly become one of the most popular platforms in the three-dimensional (3D) modeling process because of their capacity to operate in inaccessible or high-risk environments without putting the user or the researcher in harm’s way [9]. Their primary uses are in the fields of observation, monitoring, mapping, 3D modeling, and 3D reconstruction, but they also find applications in archeology, remote sensing, environmental science, geophysics, and related fields [9]. Digital maps, complex point clouds, georeferenced orthophotomaps, digital elevation models (DEMs), and digital surface models (DSMs) can all be obtained with the help of such implementations and can be used for spatial analysis and in GIS programs [10,11,12].

Regarding topics related to measurements using the most complex, vast, and versatile technology, from the point of view of construction principles, operation, and use, terrestrial laser scanning (TLS), depending on the principle used, presents different advantages [4,13,14].

Therefore, when measuring with the help of light waves, scanners based on the principle of triangulation, given the physical limitations of using a larger recording base for the camera’s field of view (FOV), are found only in near-field applications (a distance of up to 10 m), but their advantage compared with instruments that can be used at longer distances with good results, such as terrestrial laser scanners that are based on the principle of time measurement, is revealed by their much higher precision, which falls into the submillimeter range. Still, remaining in this sphere of scanning, the principle of measuring a phase difference means that the technologies developed that are based on it have a proven advantage in that the reflected signal is not influenced by different external factors, such as, for example, sunlight. However, from the perspective of the maximum precision achieved, systems based on the principle of optical interferometry represent the fiercest competition for scanners that use the principle of triangulation, but their major downside is one worth considering: high costs [4,13,14].

One of the most important applications of scanners in assessing the status of an object is the monitoring of structural deformations [15,16]. A detailed three-dimensional (3D) structure model constructed from point clouds recorded with a laser scanner is frequently used as a reference for other types of data, such as digital photogrammetry [17]. Terrestrial laser scanning (TLS) data with high resolution are often regarded as an alternative in order to visualize structural assessment and monitoring [18,19].

Remote sensing is another important aspect of TLS [20]. It makes inspection and monitoring easier without disrupting a facility’s operations or altering its structure by planting monitoring targets. It also decreases the demand for specialized employees, such as utility climbers, masons, cranes, etc., and the need to monitor specialized surveyors, as TLS can be operated by a single person [21].

On the one hand, classical methods can be described as being impractical in the case of large constructions with special architecture and, on the other hand, “subjective”, considering the limited number of data obtained from field measurements.

The replacement of single-point observations with massive datasets is essential for achieving excellent inspection and monitoring results in the matter of accuracy in obtaining a model. By using single-point observations, an overview of the deformations in the studied construction cannot be obtained with the accuracy offered by the TLS and Unmanned Aerial Systems (UAS) 3D modeling methods [9,19]. The large datasets call for the implementation of suitable methods and techniques in order to process and analyze them [22,23].

There is a clear tendency to change techniques because of technological advancements, especially in technologies used in these types of engineering measurements. One of the most important goals in this field of research is that of automation, to speed up the process as fast as possible, which will result in a decrease in measurement uncertainties by eliminating the errors caused by the human factor. The results should also lead to a decrease in the actual working time and in the costs related to each measurement operation [4,23,24,25,26,27,28].

Unmanned Aerial Systems (UASs) [9], terrestrial laser scanning (TLS) [10], laser interferometry [29], and microwave interferometry (MI) [30] are examples of applicable technologies for contactless measurements. However, by comparing all these technologies with each other and also comparing them with other classic contactless measuring technologies, TLS and UASs turn out to be the technologies that provide the best spatially distributed acquisition of structural responses. The spatial resolution of these types of sensors offers the benefit of monitoring large parts of a structure using a single sensor, enabling a more thorough comprehension of the structural reaction [31].

From the point of view of monitoring movements called oscillations, which are cataloged according to their duration as the shortest position changes in a characteristic point, two types of technologies are used. First are 3D Laser Doppler Scanning Vibrometry Systems [32], which are used only for close-range measurements. However, the method that offers the best results in the case of far-field measurements is the method that uses radar interferometry, with millimeter or even submillimeter precision under optimal conditions. The less advantageous aspects of this method are a maximum measuring distance of 1 km and the possibility of acquiring information in just one dimension (1D), but these are compensated by the advantage of monitoring a multitude of points, at frequencies up to 200 Hz, simultaneously [33].

Consequently, this research focuses on proposing a new methodology that involves the data acquired by two types of sensors, namely, the sensors used by TLS, which are typically used in 3D modeling in order to capture static environments [14,34], and the sensors used by UASs, which are now also a viable option for 3D modeling, architecture, land monitoring, and geophysical surveys [1,6,9] thanks to advancements in image capturing and processing.

### Problem Statement

By combining TLS techniques and orthophotomaps obtained with UAS, a 3D model of the Institute of Advanced Horticultural Research of Transylvania (ICHAT) building was created. In order to study the deformations and the displacements of the entire construction, over one year, point clouds were obtained from two epoch measurements and used.

The ICHAT building within the University of Agricultural Sciences and Veterinary Medicine Cluj-Napoca (UASVMCN) is a project of major importance for the university, the municipality, and for the entire north-west part of the country, being one of the largest and most modern research centers for horticulture in Romania. The building is structured as follows: a basement, the ground floor, and four upper floors.

Taking into account the fact that the construction was built through a project that was co-financed by the European Union and is still under warranty, it was not possible to mount classic monitoring targets because intervention on its resistance structure was not allowed during the warranty period. This is why a method of monitoring over time using non-destructive technologies by means of TLS and othophotomaps obtained via UASs was adopted.

Given the area where it is located and the characteristics of the foundation soil, the building is prone to the appearance of deformations and displacements, so, for this reason, special measures, such as an over-enforced foundation and underground water drainage, were considered in the design phase in order to avoid these phenomena.

Several case studies regarding the application of TLS for surface modeling and monitoring various structures [16,19,35] have addressed this issue but it has been not combined with other photogrammetric methods. Research on the application and correction of radiometric data defined using the intensity of laser beam reflections [36,37,38,39] or realistic visualizations of damage using integrating point clouds with digital photogrammetry [40] were also examined in detail in order to begin this research study. In the process of selecting the types of instruments and the right methods to use in this research, the current state of knowledge regarding UAS-based inspection and monitoring of structures was studied through a comprehensive review of recent developments [41,42,43].

Techniques for processing point clouds and analyzing the deformations in a 3D format are being developed in the direction of specific algorithm solutions [19,23,44,45,46].

By performing photogrammetric flights with a UAS, the problem of inaccessibility on some parts of the building used for this study was solved; this is the main contribution of this research. The data thus obtained were used to complete and improve the 3D models obtained with the help of TLS. Moreover, the contribution regarding both the combination of data obtained in two different ways (TLS and the UAS), as well as the use of these results for the identification of deformations in a large building without affecting its structure or facades, can be mentioned. In order to achieve these major goals, it was necessary to complete several stages, starting with the choice of the type of instruments, continuing with the elaboration of a workflow, and concretizing with the implementation of that workflow in several clearly established steps; in this way, a new methodology in the process of 3D modeling, and especially 3D monitoring, was developed.

## 2. Materials and Methods

Of the two existing and widely used paradigms, in the positivist one, reality is viewed from an objective point of view, with measurable notions and cause–effect relationships, while in the interpretivist paradigm, reality is represented by a subjective interpretation, created by each individual in his own terms and view. The two are not mutually exclusive and cannot be considered “parallel” but rather two ends of a continuum.

This study was approached from a positivist paradigm point of view, using quantitative methods, as presented in Figure 1 [47,48,49,50].

In the positivist paradigm, the truth is found via quantitative research, meaning that the phenomena are quantifiable and can be measured. Furthermore, positivist methodologies do not take into consideration the personal experience that people have, only the measurable facts [50].

One of the reasons behind the choice of this type of approach was the fact that the reliability of the results is high and can be generalized, and another reason was that a study such as this one is capable of offering an overview of the researched problem [50].

The methodology used for this research falls into the area of experimental studies. The relationship that is being studied is one between two different epochs of observations of the same object: the ICHAT building.

The problem of implementing modern measurement technologies and techniques in the positioning and monitoring of various objects and constructions, at an international level, falls into the category of activities of major importance [15,17,18,21].

Thus, in the present study, the research direction was divided into two main stages of analysis (Figure 2), namely, 3D modeling, for which different types of techniques and technologies were used, both in terms of data acquisition and their processing, representation, and interpretation, and the second part of the research, which is represented by the 3D monitoring of the building.

### 2.1. Preliminary Operations for Data Acquisition

In carrying out this first step, particularly important for subsequent works, two methods were used in order to create the support network, that is, the modeling/monitoring network.

#### 2.1.1. GNSS Measurements for the Support Network and Modeling/Monitoring Network

In the first phase, with the help of GNSS technology, two points that were integrated into the geodetic support network within the UASVMCN campus were determined using the static method; these points were later used to create the monitoring geodetic network. The GNSS receiver model with which we performed these determinations was Trimble R10, which has a planimetric accuracy of 0.008 m and a vertical accuracy of 0.010 m [51].

Also at this stage, although the model of the receiver provides real-time differential corrections without being connected by radio waves or GSM to a permanent station or base, additional corrections from the Romanian Position Determination System (ROMPOS) permanent station in Cluj-Napoca were included in the data obtained through post-processing. In this way, many causes of errors in the measurements were prevented (e.g., multipath errors). For Precise Point Positioning (PPP,) the university’s own permanent station, which is included in a national positioning network called PIVOT GNSS POS, was used.

The first determinations using this technique took place in 2020 and the second in 2021. Following the measurements and their processing, the coordinates values presented in Table 1 were obtained:

#### 2.1.2. The Use of Total Stations in Order to Create the Modeling/Monitoring Geodetic Network

Starting with the information retrieved and processed with the help of GNSS determinations (S19, S40), the modeling/monitoring geodetic network was created using the closed traverse method combined with the polar coordinates method.

The modeling/monitoring geodetic network was materialized on the ground, as seen in Figure 3 (yellow—the traverse; orange—resections; green—side shots).

### 2.2. Terrestrial Laser Scanning

#### 2.2.1. Field Operations

Leica ScanStation C10 was the instrument used for this project. Leica ScanStation C10 is a panoramic scanner, with a 360° × 270° field of view (FOW). It presents a high scan speed (up to 50,000 points/second) and very good accuracy using a spinning mirror and a high-resolution camera; the camera is helpful in texture mapping point clouds [52].

The first scan of the exterior of the ICHAT building was performed in December 2021, and the second operation of the same kind took place in December 2022.

The most well-known methods of data acquisition are registration, carried out via the free-stationing technique with the help of the three-dimensional intersection of the sights to the scanned targets; the determination of each station point via resection; registration executed by stationing at points of known coordinates; or registration using constraint points in order to perform the registration of the point clouds [10].

Considering the modeling/monitoring network previously presented and the methods also presented before, the first option was chosen for the following reasons.

The first of these reasons was related to the difficulty encountered in centering the instrument on known coordinate points, which appeared because of the heavy weight of the instrument.

The second reason was related to the next step of the work, namely, establishing the points from where the scans will be made. In this case, depending on the reflectivity of the scanned materials, the maximum measurement distance differs.

The third element that could affect the measurements in certain circumstances, besides reflectivity and distance, is the angle of incidence.

Taking into account all three constraints, the distance range at which each scanning station was positioned was at a maximum of 30–40 m away from the building because the reflectivity of the brown, gray, and black materials of the construction was very low. This kind of positioning also solved the problem of the laser beam’s angle of incidence over the building.

Once the details related to the design of the measurements were established, the acquisition of data started.

At each station point of the two measurement epochs, the same parameters for instrument settings (resolution, viewing angle, distance filters, etc.) were entered. The scans were performed by the same operator, approximately under similar atmospheric conditions (temperature, pressure, clear atmosphere).

#### 2.2.2. Processing Data Obtained from the Scans

Leica Cyclone consists of several modules, which help either with registration and georeferencing or with the creation of three-dimensional model deliverables, animations, and videos [53].

Given the chosen measurement technique, automatic registration was performed. Automatic registration is the process of combining or merging the results taken from different stations and transforming them in such a way that they all belong to the same coordinate system, automatically, without the need for intermediate steps [54,55]. From this point of view, it can be said that registration is practical the georeferencing and vice versa [54,55].

Following registration, the point clouds related to the first and second epochs of the measurements were obtained, as can be seen in Table 2 and Table 3.

One of the most important steps for obtaining a point cloud used in construction monitoring is the filtering (cleaning) of the raw point clouds (which present “noise”), as can be seen in Figure 4a,b. The filtering of the point clouds was carried out for both measurement epochs in the next steps.

### 2.3. Photogrammetric Measurements

#### 2.3.1. Field Operations

Both sets of data were obtained with the help of the UAS DJI Phantom 4 Pro RTK. The main technical specifications of the drone used are presented in Table 4; full details are available on the producer’s website [56].

The MapPilot Pro software [57] was used to create the flight plan in this photogrammetric data retrieval stage.

A single flight plan was created to avoid the occurrence of errors caused by differences in angle, altitude, resolution, or overlap. For these reasons, as Oniga V.E. et al. suggest [58], a double grid mission was planned, but for better image overlap and overall accuracy, it was also necessary to take some frames manually.

The predefined mission was saved before the first flight and repeated for the second flight. In the present case, the parameters used in its planning were the ones presented in Figure 5. Therefore, the chosen type of mission was a “grid”, and the flight altitude was set at 60 m. An 80% overlap of the frames on the longitudinal flight lines was introduced to the flight plan, and a 75% overlap was introduced for the overlap of the frames on the transversal flight lines. Maximum flight speed was set to 4.8 m/s, and the expected ground sample distance at ground level (GSD) was 0.016 m/px.

The orange dots and yellow lines represent the flight mission limits; the purple dot represents the “home point”; the white lines represent the proposed strip model; and the orange thin line represents the actual flight.

As can be seen in Figure 5, the drone completely respected the flight plan (route) in the area of interest, which is represented by the polygon defined by the yellow lines. The only areas where the aircraft deviated from the predetermined routes were places where it changed its flight direction, which were outside the area of interest. The deviations from the planned routes appeared because of the settings made during the flight planning stage; the software provides two options for changing the direction (a sudden change in direction at a right angle or a change in direction in a curve) in order to optimize the flight time.

Regarding “return to home” after completing the flight mission, this operation was performed manually to avoid some obstacles. That is the reason why the respective route is the only nonlinear one, as can be seen in Figure 5.

The actual UAS measurements took approximately two hours each, taking into account the acclimatization and installation of the equipment.

#### 2.3.2. Photogrammetric Data Processing

The first software used was Agisoft Metashape, a program specialized in 3D modeling and generating orthophotomaps based on digital frames.

In order to create a 3D model using photogrammetric methods, a series of steps must be completed [58].

First, a project must be created; then, the raw images will be imported into that project.

Subsequently, a coordinate system will have to be chosen; in this case, the Stereo70—Pulkovo 1942 was the choice.

The next step involved aligning the frames and generating the connection point cloud and the point cloud, which is generated based on that alignment.

The insertion and identification of control points on the frames and adjusting their ground control points (GCPs), thus establishing their determination accuracy, are among the most important steps of this stage.

The last phase involved generating a dense point cloud and the 3D model.

As an option to evaluate the uncertainties regarding the results of the photogrammetric data, the reprojection errors of tie points are presented in Table 5 as a root-mean-square error (RMSE) determined based on the ground control points (GCPs—X, Y, Z errors). In the last column of Table 5, the reprojection errors in the pixels are also presented, together with the number of projections for a GCP location [59].

### 2.4. Data Processing in Order to Merge the Products Obtained Using the Two Methods Presented before and Creating the 3D Model and the Comparative Model

#### 2.4.1. Main Reason for the Necessity of Merging

Because this building is the headquarters of the Horticulture Faculty of the university, on some of the terraces, different kinds of flora are exposed. This consists of a 20–50 cm layer of soil, plus different plants or lawns, or even gravel in some places; this type of building is considered a beneficial option in terms of thermal efficiency. That is why, to achieve reliable results, two different technologies, scanning and photogrammetry, were used.

Thus, the problem that occurred in the case of the terrestrial laser scanner being stationed on the terraces was the existence of several hydroprotective, fireproof, and thermoprotective films deployed under the green layers (soil/plants) or under the gravel, which, if damaged, would have caused damage inside the construction (mostly infiltrations, heat leaks, or even fires).

#### 2.4.2. Filtering/Cleaning

In order to carry out the processing work, the Global Mapper software was chosen, which is a software package that can be successfully used to visualize and manipulate point clouds or filter data based on height.

Furthermore, the imported data were filtered based on longitudinal, transversal, or vertical profiles; this action was of particular importance in the present case.

After importing the point clouds into Global Mapper, they were “cleaned” in order to remove data that were not conclusive for this study as seen in another part of this paper.

After the rough cleaning of the point clouds, another important step was to separate the point clouds according to the area of interest, namely, processing the construction separately from that of the surrounding environment, of which the exterior stairs were considered a part.

The next stage involved filtering the dense point clouds obtained via photogrammetric methods according to the point elevations.

For this, the point clouds were inserted one by one, keeping only the information contained in certain previously chosen elevation ranges; in this way, the filtering was much improved, and the actual work time was much shortened compared with the classic data processing mode (manual selection).

In the last step of filtering and “cleaning”, the point clouds were executed using the longitudinal, vertical, and cross-sections (profiles), as presented in Figure 6.

The possibility of both 2D and 3D visualizations of the same selection or profile not only facilitates but also improves the filtering of point clouds by eliminating noise and useless data, thus obtaining the final construction elements of interest, namely, the terraces and the ceilings.

## 3. Results

### 3.1. Results Related to Terrestrial Laser Scanning

After filtering the point clouds from the two scan stages, the data were exported from Leica Cyclone in *.PTS format in order to be used in the following stages.

The results obtained after the filtering of the point clouds related to terrestrial laser scanning are presented in Figure 7.

Even if the initial intention was to keep the meteorological and atmospheric conditions as similar as possible, small differences can nonetheless be observed in the point clouds obtained during the two measurement epochs, with differences caused by atmospheric conditions.

Even if there are differences in the point clouds, they mostly influence only the recorded RGB (color) code, not the density of the obtained point cloud, implicitly not affecting the accuracy of the data.

### 3.2. Results Related to Photogrammetric Measurements

The main challenge was the fact that the studied building has mostly glazed facades, a particularly difficult material to model using photogrammetric methods and terrestrial laser scanning technology.

It should be noted that the data acquisition, both photogrammetrically and via terrestrial laser scanning, was carried out in well-established periods before the glazed facades of the building were cleaned. This option was chosen because, at that moment, many impurities have deposited in the glazed surfaces of the building, which causes the transparency to be greatly reduced.

The dense point cloud obtained via photogrammetric methods presented in Figure 8 was the product used to further survey ceilings and terraces that were inaccessible to terrestrial laser scanning.

Based on the dense point cloud seen in Figure 8 (left), a grid consisting of a total of 38,643,799 polygons, presented in Figure 8 (right), was additionally generated, thus also rendering a 3D model of the studied building.

### 3.3. Results Related to the Merging of Datasets

#### 3.3.1. Terraces and Ceilings Extracted from the Photogrammetric Dataset

Considering the same point cloud had to be imported several times in the same project because of the chosen filtering mode (based on the elevations), several layers were basically created, which is a difficult feat to manage. To solve this inconvenience, these layers were grouped into one by exporting them as a single point cloud; the results for the first dataset are presented in Figure 9.

#### 3.3.2. Final 3D Model and Comparative Model

Because both the registration and the georeferencing of the point clouds obtained by terrestrial laser scanning and the processing of the photogrammetric data were carried out using the same coordinate system (Stereo70—Pulkovo 1942), overlap between the two filtered datasets was achieved automatically (Figure 10).

The 3D model and, implicitly, the point clouds corresponding to the entire ICHAT building of the UASVMCN campus were obtained after following the steps mentioned earlier. In Figure 10 (left) an overview of the point cloud obtained is presented, and a detailed view of the 3D model can be seen in Figure 10 (right).

#### 3.3.3. Detail Analysis

Figure 11, Figure 12 and Figure 13 show 3D captures of the same metal design elements of the building and part of its highest terrace. In the case of the metallic elements captured in these images, the impact that the combination of the two techniques, that is, the data acquisition/processing technologies, had on the final 3D model of the object can be best observed.

In Figure 11, which represents the merging of the datasets in Global Mapper, the metal elements highlighted in dark green illustrate the parts of the object surveyed by terrestrial laser scanning technology, and the same metal elements represented in gray illustrate the parts of the object investigated via photogrammetric methods.

The complete model was later imported into Leica Cyclone where it was studied from other points of view.

Using Leica Cyclone, the results were visualized according to the RGB codes of each point in the point cloud, as seen in Figure 12, and according to the intensity of the reflected laser beam, as presented in Figure 13, which provides a much better perspective on the working precision and on the differences between the techniques and technologies.

In the case of manipulating the combined datasets in Leica Cyclone, represented by the intensity of the reflected laser beam point of view, the points retrieved by means of terrestrial laser scanning are illustrated in red, while the points retrieved with the help of UAS photogrammetric technologies are illustrated in shades of yellow, green, and blue (Figure 14).

### 3.4. Deformation Analysis for the Studied Building

#### 3.4.1. CloudCompare Operations

To carry out this study, the CloudCompare software was used, which—in addition to similar facilities to the other point cloud processing and manipulation software presented so far—presents some particularities.

Among these particularities, the one that represented the greatest interest was the possibility of comparing two or more datasets based on various parameters.

In this sense, the products resulting from the first measurement epoch, together with those obtained from the second measurement epoch, were imported into the same project using CloudCompare.

Using the “Cloud-to-Cloud Absolute Distance” [60] command, the distances between the two point clouds were automatically calculated.

Following the execution of this command, based on the results obtained with the help of the calculation program, the maximum distance between some points of the datasets (negligible noise points omitted during filtering) was found to be 1 m, but the average of the positioning differences fell within a range of 0–0.03 m.

Consequently, in order for the results of this operation to be as correct as possible and, at the same time, interpreted as easily as possible, a common point cloud in which differences in positioning with a distance of up to 0.03 m are highlighted was extracted; this representation is illustrated using a blue–green–yellow–red color scale, scored automatically by the software based on the density of each interval.

#### 3.4.2. Main Findings

After determining the deformations and displacements of the monitored building point of view, the results are as follows:In Figure 15, it can be seen that the northern facade presents the largest deformations, which are also present on the largest surfaces of all the facades. They are in a range of 0.003–0.03 m, illustrated based on the color scale. From the dispersion point of view, the largest distances between the two datasets in this facade are found in the upper part of the building and, especially, the eastern part of it, which is covered in aluminum bond (a construction material used to cover facades).

Figure 16 and Figure 17 highlight the fact that the western part of the building shows great stability, especially in the southern area. The values reaching 0.016–0.018 m are only at the upper levels of the northern part of the façade; otherwise, they fall within a range of 0–0.015 m.

**Figure 16 sensors-23-05678-f016:**
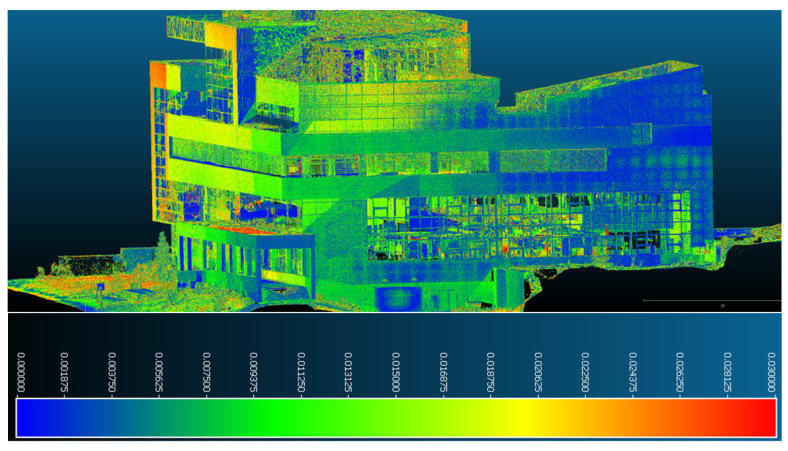
Point-to-point displacements/deformations—western façade from the northern perspective.

**Figure 17 sensors-23-05678-f017:**
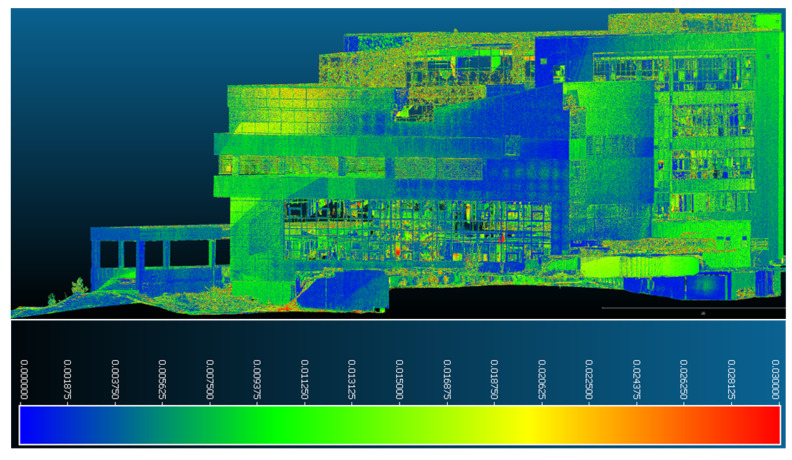
Point-to-point displacements/deformations—western façade from the southern perspective.

In the case of the southern facade of the studied building presented in Figure 18 and Figure 19, displacements and deformations determined by comparing the two point clouds taken approximately one year apart were found in its extremities, especially in the upper part, and the distance interval included was between 0–0.011 m. In the central area, consistent stability was observed with values not exceeding 0.003–0.005 m.

**Figure 18 sensors-23-05678-f018:**
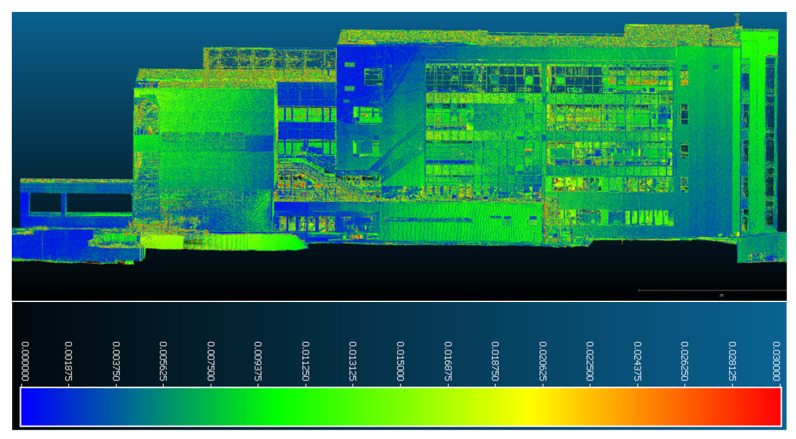
Point-to-point displacements/deformations—southern facade.

**Figure 19 sensors-23-05678-f019:**
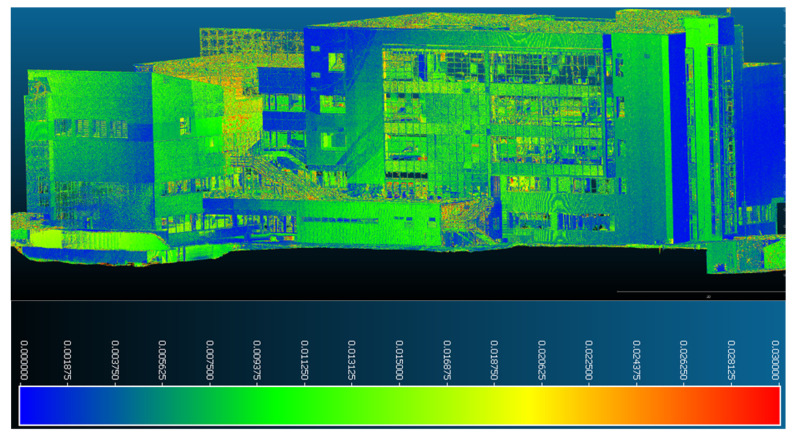
Point-to-point displacements/deformations—southern facade from the eastern perspective.

The eastern facade seen in Figure 20 is the most stable component of the building, with values that do not exceed 0.002–0.003 m along the entire length of the facade.

**Figure 20 sensors-23-05678-f020:**
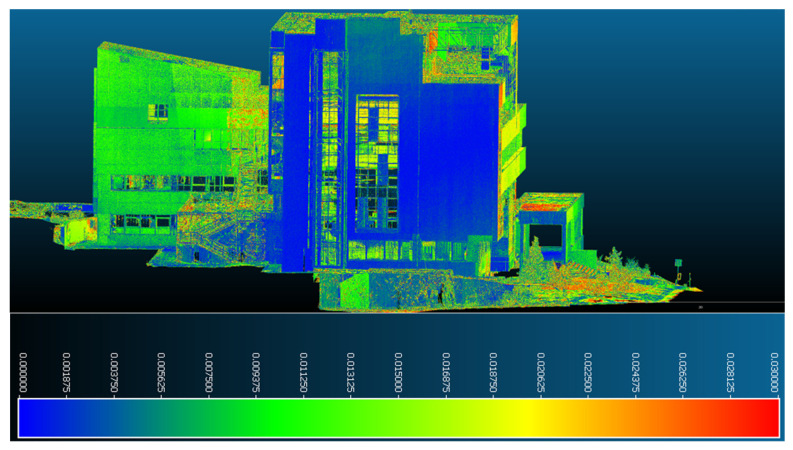
Point-to-point displacements/deformations—eastern facade.

In Figure 21, having also noted the results regarding the other facades, the points derived from the point cloud obtained via photogrammetric methods can easily be seen, and they mostly represent terraces covered with vegetation or decorative gravel (which is sensitive to any change in temperature or precipitation); these represent the largest and most obvious displacements and deformations. For this reason, they were not taken into consideration when drawing up conclusions on the stability of the studied construction.

This consideration is highlighted in Figure 21 by illustrating a transversal section (profile) of one of the terraces modeled according to the previous descriptions; by merging the data obtained with the help of the two different methods, the positional changes in the points describing the vegetation layer can be clearly observed compared with those that represent the fixed elements of the building.

Although the detailed analysis of the displacements and deformations of the ICHAT building showed the existence, in the case of certain areas, of significant changes in position and shape, based on Figure 22—in which the histogram related to the number of points in the comparative measurement located at a certain distance from those in the reference measurement is illustrated—it can be stated that the volume of points with displacements greater than 0.015 m is significantly lower compared with that of points with displacements smaller than this value.

Thus, the construction can be considered stable, but monitoring works must be continued for several more years given the relatively short time since the construction was completed.

## 4. Conclusions

As mentioned, measurements for this case study were carried out in four stages between 2020 and 2022.

The terrestrial laser scanning and aerial photogrammetry work was managed separately during all stages, and the data processing was conducted separately only in the first phase (TLS: point cloud registration, noise filtering, extraction of point cloud related to the studied area; UAS: photogrammetric processing, noise filtering, extraction of point cloud related to the studied area); later, all analyses, representations, and interpretations were carried out using the combined point cloud.

Between the two point cloud manipulation programs used in this work, it was found that the results of the merged point clouds obtained via two completely different methods, that is, TLS and UAS photogrammetric flights, were most efficiently highlighted with the help of the Leica Cyclone software.

Determining the deformations and displacements in the construction in the studied time interval was carried out using the CloudCompare software. In this sense, although the occurrence of additional errors in the values of displacements and deformations in the areas of the building wrapped in aluminum bond (a material relatively sensitive to temperature differences) was expected, given that the series of measurements were carried out according to work plans—which were imposed to carry out the work in approximately the same environmental conditions—changes in the position and shape of the studied building were found to be randomly dispersed over its entire surface; thus, in this way, it can be stated that the covering material of the structure, whether it was aluminum bond or classic thermal insulation (with decorative plaster), did not significantly influence the final results.

The methods studied in this article are appropriate to use, especially for buildings that require monitoring over time using non-invasive methods. Furthermore, these methods allow for the tracking of deformations in the whole built assembly, unlike the classic methods, which only offer a single-point-based representation of these deformations.

In the end, the studied methodology proved to be particularly useful from several points of view, but the most important aspect to mention is the fact that the main objective of this work was fulfilled. More precisely, these non-destructive measurements for modeling and, especially, monitoring the behavior of large constructions in a 3D format over time can be successfully used in current practices with good results and accuracy that falls within the required tolerances usually asked for in this type of engineering work.

### 4.1. Recommendations

As a first recommendation, it is important to perform measurements in the same meteorological and atmospheric conditions.

We also recommend using the same geodetic network (reference points, ground control points, and checkpoints) and verifying the stability of the geodetic network from one stage to another.

Another recommendation is to apply the same flight plan within all epochs of photogrammetric measurements in order to be sure that the 3D model obtained is reconstructed from images that keep the same characteristics (position, yaw, pitch, roll) from one epoch to another.

### 4.2. Strengths and Limitations

This study uses mixed methods to retrieve, process, and analyze the data. The advantage of this approach is the large volume of information retrieved in a relatively short period of time with great accuracy and which, in the end, led to very good results that can be used in various other studies.

This study has limitations besides the ones that are common to the research paradigm approached. Gathering samples used for the quantitative analysis of data regarding the principle of merging the terrestrial laser scanning process with photogrammetrically obtained data from large constructions rather than using those methods independently is a process that can be improved for further research in the areas of precision and accuracy.

### 4.3. Future Research

The studies carried out in this work can be continued and developed.

Some of the research perspectives include continuing the monitoring of the studied construction through scans combined with annual photogrammetric flights and trying to simplify and automate both the data acquisition process and data processing even more.

Using different technologies to retrieve information, such as LiDAR or aerial laser scanning, can also be considered a point of interest for future research work.

Scanning the exteriors and the interiors of the rest of the buildings in order to integrate them into a future interactive 3D map of the UASVMCN campus is an ambitious idea for better and faster integration of freshmen into campus student life.

## Figures and Tables

**Figure 1 sensors-23-05678-f001:**
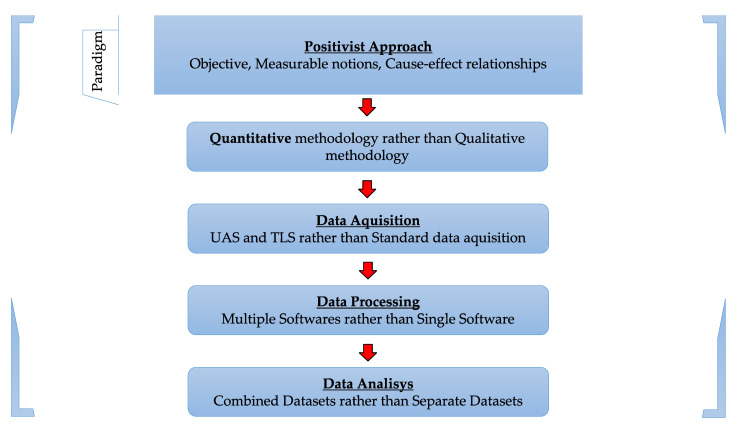
Research paradigm [50].

**Figure 2 sensors-23-05678-f002:**
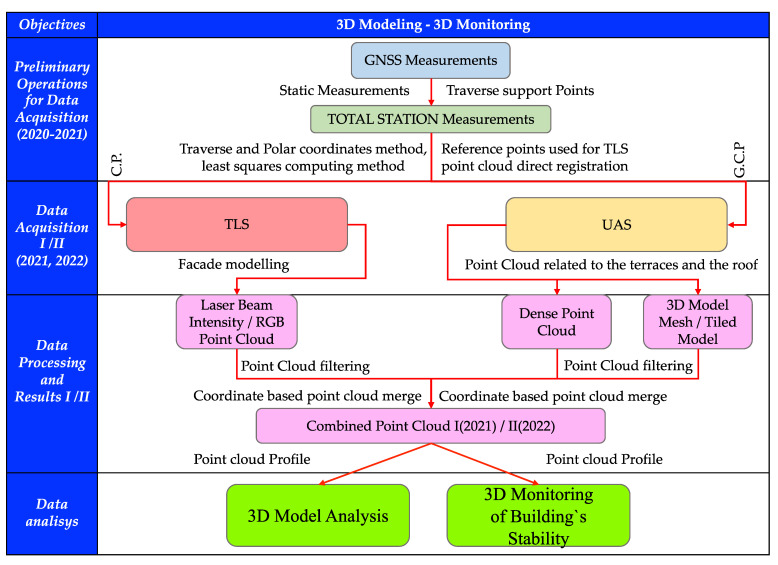
Proposed workflow for 3D modeling and monitoring [10].

**Figure 3 sensors-23-05678-f003:**
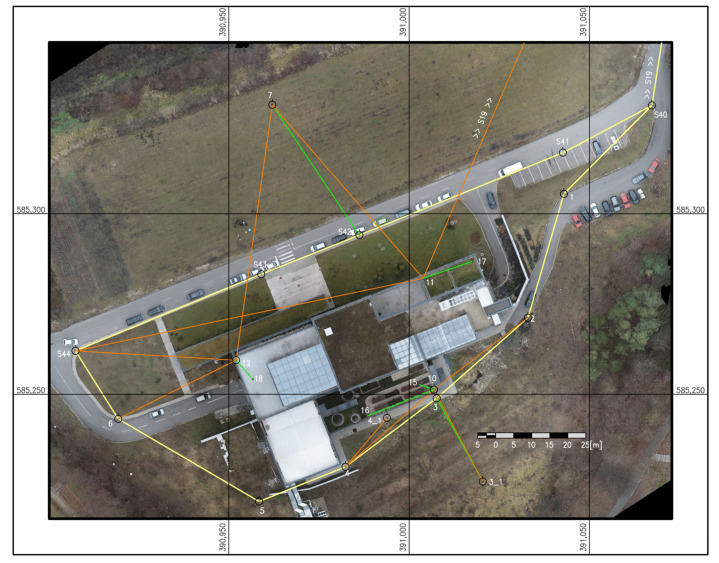
Modeling/monitoring network.

**Figure 4 sensors-23-05678-f004:**
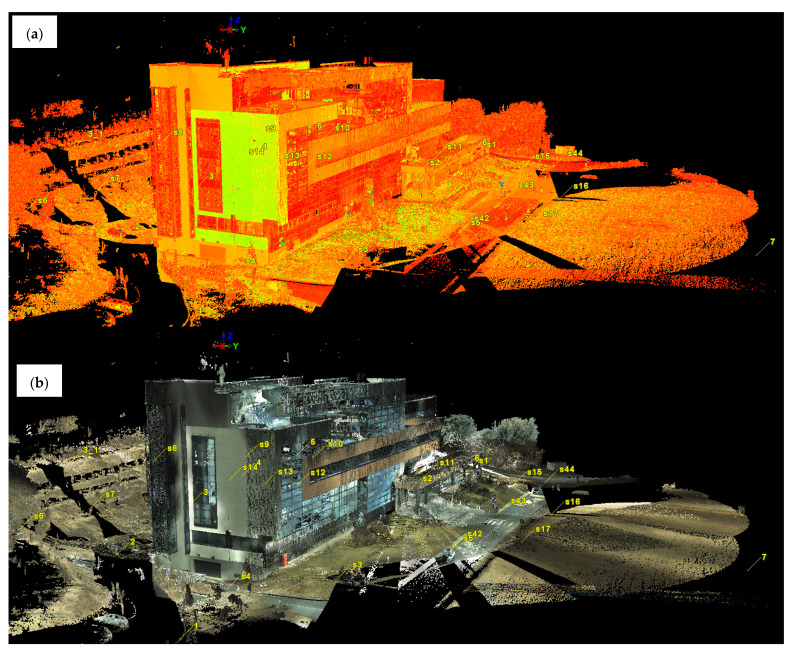
(**a**) The raw point cloud obtained from the registration of the first epoch of measurements (visualization based on the intensity of the reflected laser beam). (**b**) The raw point cloud obtained from the registration of the first epoch of measurements (view based on the RGB code taken from the images).

**Figure 5 sensors-23-05678-f005:**
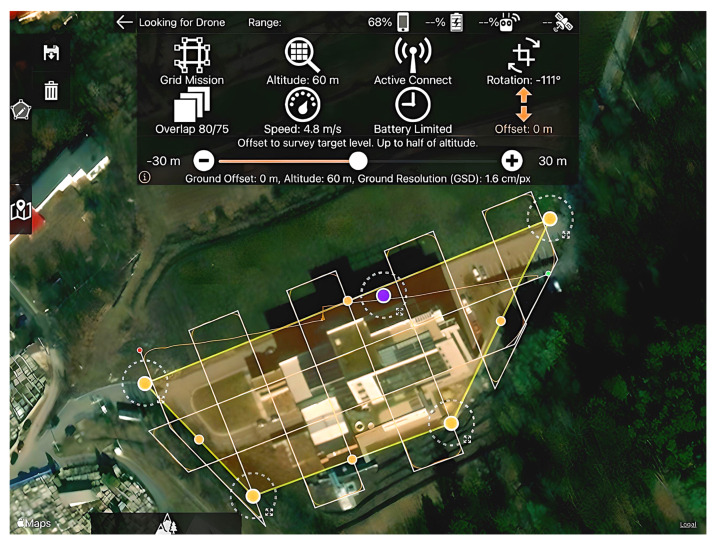
Flight plan—screen capture from Map Pilot Pro.

**Figure 6 sensors-23-05678-f006:**
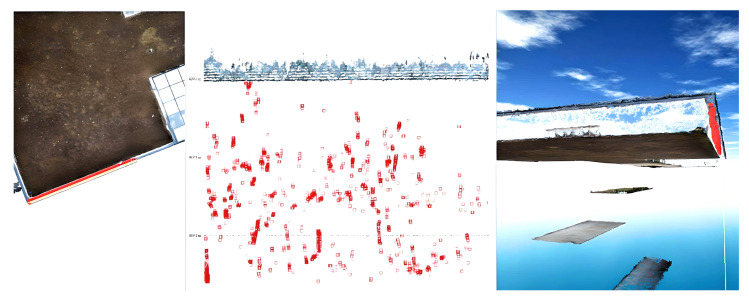
Detailed filtering of the point cloud using profiles.

**Figure 7 sensors-23-05678-f007:**
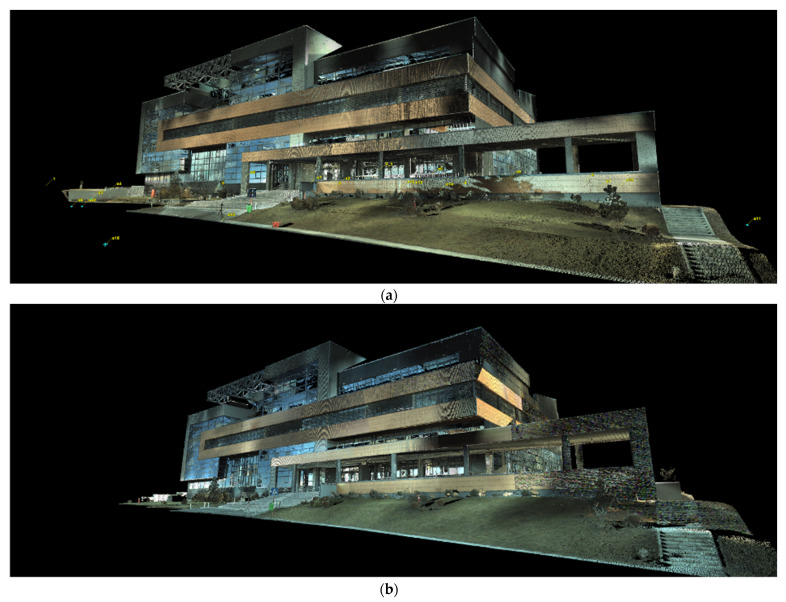
(**a**) Point cloud obtained from terrestrial laser scanning in the first epoch of measurements filtered within Leica Cyclone (RGB code taken from the images). (**b**) Point cloud obtained from terrestrial laser scanning in the second epoch of measurements filtered within Leica Cyclone (RGB code taken from the images).

**Figure 8 sensors-23-05678-f008:**
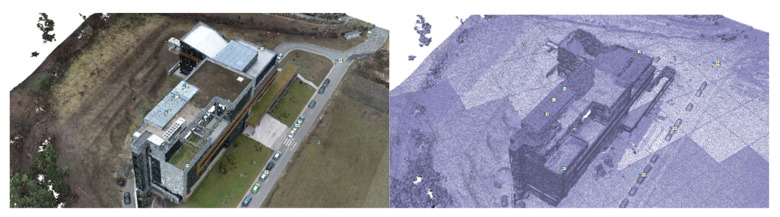
Dense point cloud (**left**) and TIN network overview (**right**).

**Figure 9 sensors-23-05678-f009:**
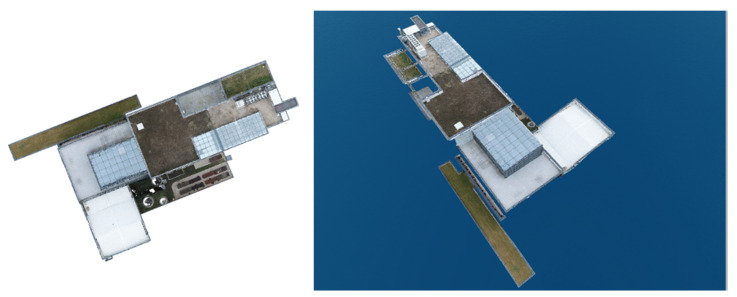
The results obtained after filtering the dense point cloud based on altitudes (terraces and ceilings).

**Figure 10 sensors-23-05678-f010:**
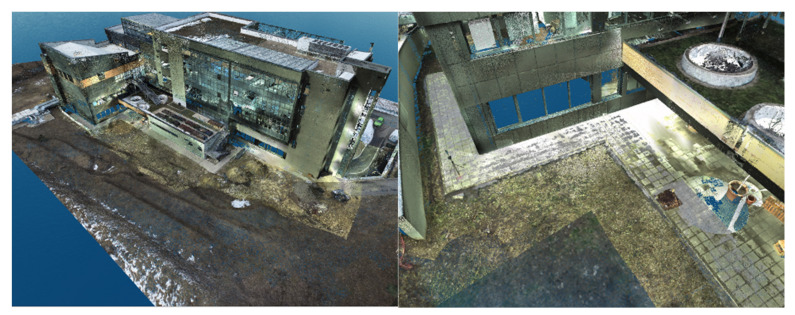
Overview of the 3D model (**left**) and a detailed view of the 3D model (**right**).

**Figure 11 sensors-23-05678-f011:**
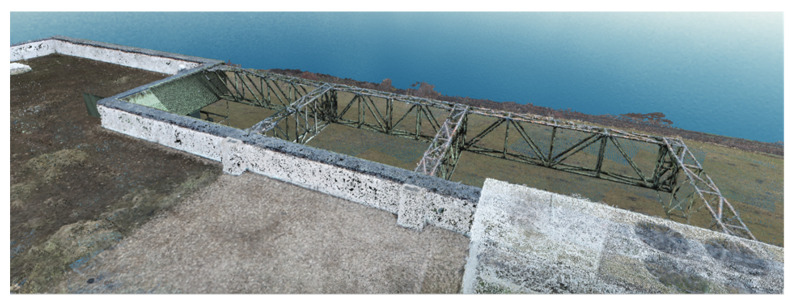
Metal elements of the merged dataset—Global Mapper.

**Figure 12 sensors-23-05678-f012:**
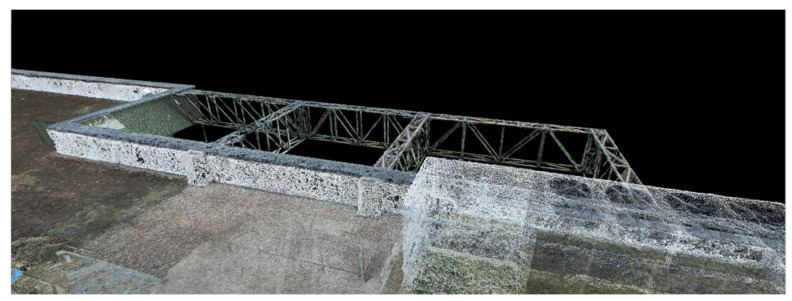
Metal elements from the merged dataset—Leica Cyclone—view based on the RGB code taken from the images.

**Figure 13 sensors-23-05678-f013:**
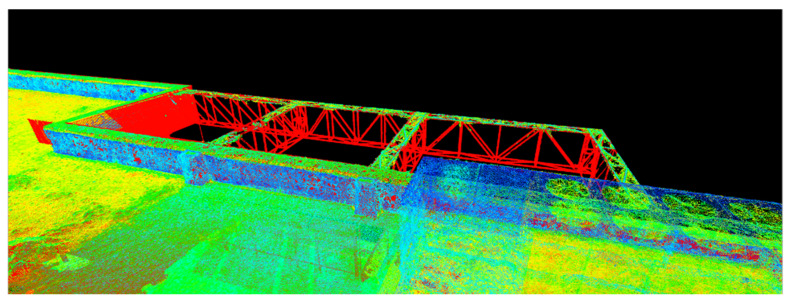
Metal elements from the merged dataset—Leica Cyclone—view based on the intensity of the reflected laser beam.

**Figure 14 sensors-23-05678-f014:**
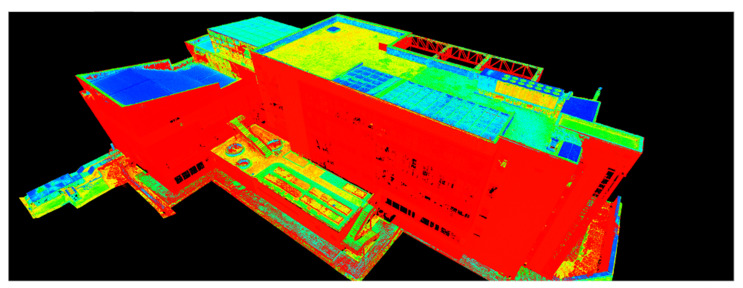
Overview of the ICHAT building from the merged dataset—Leica Cyclone—view based on the intensity of the reflected laser beam.

**Figure 15 sensors-23-05678-f015:**
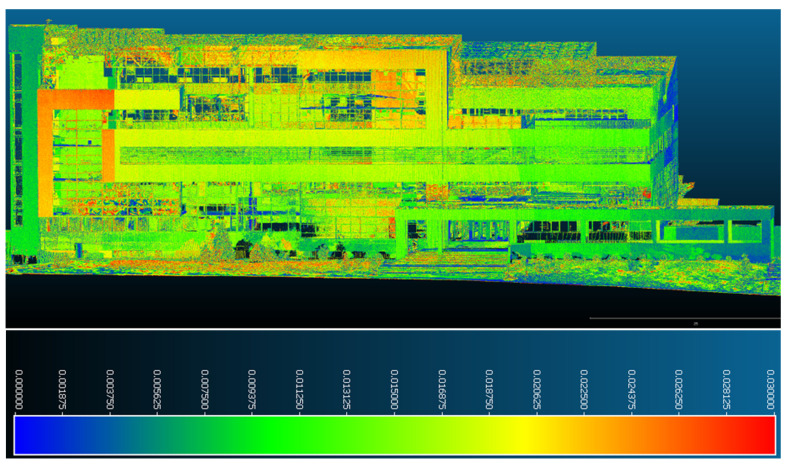
Point-to-point displacements/deformations—northern facade.

**Figure 21 sensors-23-05678-f021:**
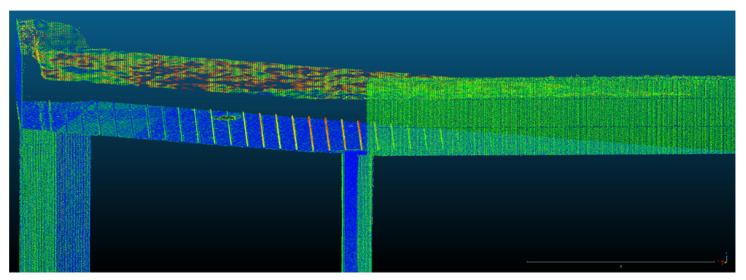
Detailed view of a cross-section.

**Figure 22 sensors-23-05678-f022:**
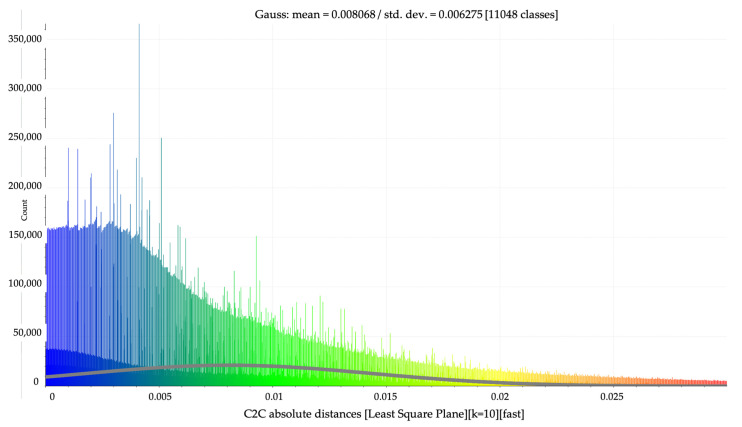
Histogram: point volume—displacement range.

**Table 1 sensors-23-05678-t001:** S19 and S40 coordinates (2020 and 2021 and the results of the arithmetic mean, which were used further in this study).

Year	Coordinates Type (EPSG 3844)	S19	S40
2020	X [m]	585,486.082	585,330.031
	Y [m]	391,091.706	391,067.304
	Z [m]	378.332	395.199
2021	X [m]	585,486.089	585,330.040
	Y [m]	391,091.713	391,067.295
	Z [m]	378.343	395.207
Average	X [m]	585,486.086	585,330.036
	Y [m]	391,091.710	391,067.300
	Z [m]	378.338	395.203

**Table 2 sensors-23-05678-t002:** Registration for the first scan.

Station No.	Measurement Epoch	Scan/Points Used for Registration	Scanned Area
1	I	Scan 1/Resection 5, 6, S44	2356 × 6282 points
2	I	Scan 2/Resection 4, 5, 6	4363 × 4398 points
3	I	Scan 3/Resection 5, 6, S44	4363 × 5060 points
4	I	Scan 4/Resection 3_1, 4	2356 × 6282 points
5	I	Scan 5/Resection 3, 3_1, 4	2356 × 6282 points
6	I	Scan 6/Resection 4, 3, 3_1	2268 × 2442 points
7	I	Scan 7/Resection 6, S44, S43	4363 × 3246 points
8	I	Scan 8/Resection 7, S43, S44	4363 × 4712 points
9	I	Scan 9/Resection 7, S42, S43	4363 × 4712 points
10	I	Scan 10/Resection S43, S42, 7	2356 × 6282 points
11	I	Scan 11/Resection S41, 7, S42	4537 × 6876 points
12	I	Scan 12 Resection 1, 7, S42	4537 × 4258 points
13	I	Scan 13/Resection 2, 3	4363 × 1918 points
14	I	Scan 14/Resection 3, 2	4363 × 3246 points
15	I	Scan 15/Resection 4, 3, 3_1	4363 × 4536 points
16	I	Scan 16/Resection 5, 4 3	2268 × 2042 points
Total			164,184,441 points

**Table 3 sensors-23-05678-t003:** Registration for the second scan.

Station No.	Measurement Epoch	Scan/Points Used for Registration	Scanned Area
1	II	Scan 1/Resection 5, 6, S44	2356 × 6282 points
2	II	Scan 2/Resection S43, S42, 7	4363 × 4398 points
3	II	Scan 3/Resection 3_1, 4	2356 × 6282 points
4	II	Scan 4/Resection S43, 7, S44	4014 × 4188 points
5	II	Scan 5/Resection S43, S42, 7	4363 × 4188 points
6	II	Scan 6/Resection S41, S42, 7	4363 × 4712 points
7	II	Scan 7/Resection S42, 1, S41	4363 × 5060 points
8	II	Scan 8/Resection S42, S41, 2	4363 × 3490 points
9	II	Scan 9/Resection 3, 2, 1	4014 × 2792 points
10	II	Scan 10/Resection 3, 4, 3_1	4014 × 4712 points
11	II	Scan 11/Resection 3_1, 3_4	4537 × 4886 points
12	II	Scan 12 Resection 6, 5, 4	4363 × 3490 points
13	II	Scan 13/Resection 5, 6	4363 × 5410 points
14	II	Scan 14/Resection 4, 3_1, 3	4537 × 6632 points
15	II	Scan 15/Resection 5, 6, S44	4537 × 6980 points
16	II	Scan 16/Resection 7, S44, 6	4014 × 1570 points
Total			175,931,341 points

**Table 4 sensors-23-05678-t004:** Technical specifications of the DJI Phantom 4 RTK drone [56].

Component	Specification	Value
Aircraft	Max Service Ceiling Above Sea Level	6000 m
	Max Speed	31 mph (50 kmph) (P-mode) 36 mph (58 kmph) (A-mode)
	Max Flight Time	Approx. 30 min
	Operating Temperature Range	32° to 104° F (0° to 40° C)
	Hover Accuracy Range	RTK enabled and functioning properly: Vertical: ±0.1 m; Horizontal: ±0.1 m RTK disabled ±1.5 m (with GNSS positioning)
	Image Position Offset	The position of the camera center is relative to the phase center of the onboard D-RTK antenna under the aircraft body’s axis (36, 0, and 192 mm), applied to the image coordinates in Exif data.
Camera	Sensor	1″ CMOS; effective pixels: 20 M
	Lens	FOV 84°; 8.8 mm/24 mm (35 mm format equivalent: 24 mm); f/2.8–f/11, autofocus at 1 m–∞
	Mechanical Shutter Speed	8–1/2000 s
	Electronic Shutter Speed	8–1/2000 s
	Max Image Size	4864 × 3648 (4:3) 5472 × 3648 (3:2)
GNSS	Multi-Frequency Multi-System High-Precision RTK GNSS	Frequency used: GPS: L1/L2; GLONASS: L1/L2; BeiDou: B1/B2; Galileo: E1/E5a First-fixed time: <50 s Positioning accuracy: Vertical 0.015 m + 1 ppm (RMS); Horizontal 0.01 m + 1 ppm (RMS) 1 ppm means the error has a 1 mm increase for every 1 km of movement from the aircraft.

**Table 5 sensors-23-05678-t005:** Root-mean-square error (RMSE) for tie points determined based on the ground control points.

Label	X Error (m)	Y Error (m)	Z Error (m)	Total (m)	Image (pix)
15	0.00058173	0.00125755	0.00581661	0.00597937	0.00259 (15)
17	0.00548431	−0.0060989	−0.00235188	0.00655962	0.00152 (13)
S40	0.000131755	0.00269615	0.000280376	0.00271389	0.00086 (4)
4_1	−0.00153069	−0.000534017	−0.0026875	0.0031386	0.00163 (4)
S44	0.00608087	−0.00177402	0.00252231	0.00681808	0.00263 (7)
S43	−0.0132488	0.00698395	−0.00443419	0.0156195	0.00211 (10)
S42	0.00741983	−0.0023309	0.000796942	0.00781806	0.00153 (9)
Total	0.00621705	0.00384871	0.00323651	0.0079962	0.00204

## Data Availability

The data presented in this study are available on request from the corresponding author. The data are not publicly available due to high volume of data caused by the increased point density.

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
