# Peer review of "Non-Destructive Measurements for 3D Modeling and Monitoring of Large Buildings Using Terrestrial Laser Scanning and Unmanned Aerial Systems"

_sensors, 2023, doi:10.3390/s23125678_

Round 1

Reviewer 1 Report

Dear authors,

Below please find some remarks considering your paper.

·         In the case of the title and the abstract, those elements must be understandable to none familiarise with the field reader. It is advised to use no abbreviations if possible. In the abstract, the authors are using abbreviations, which should be avoided in the abstract and those abbreviations should be introduced for the first time in the main text. Otherwise, the abstract is good.

·         The reference list is good with some minor adjustments to be made in case of missing state-of-the-art elements – look next point.

·         The introduction part is written clearly. However:

o   The introduction needs to be reconsidered in the case of the structure. Some basic information on the field are given at the end eg. Lines 59-60 and, especially, the beginning of the introduction is abrupt. The authors basically go straight to  Terrestrial Laser Scanning with no description of the field. The general description/introduction to the field of measurements, especially, the non-destructive and with laser systems one is missing. Suggest to include why to perform such measurements, and what else are non-destructive laser techniques used for e.g not only deformation but techniques like 3D Laser Doppler Vibrometry for modal analysis DOI: 10.3390/s23031263. Some other techniques are presented in lines 59-60 but with no information that those techniques are mainly for deformation monitoring.

o   The last paragraph where the aim, goal and especially the novelty of the study is to be presented is to be rewritten. These elements are presented but novelty is not clearly presented. Suggest improving.

·         Material and methods, results:

o   Fig.1 seems too big

o   Be careful about the units. For clarity remember to make a space between the value and the unit (except deg and %) eg. Error in tab1. Please check the whole article carefully. Would also suggest to, if possible, to unify the units. E.g in Table 1 in the same row mm, cm, and m are used. For clarity, it would be beneficiary to stick e.g. to SI units.

o   Fig. 22 is extremally important but details are hardly visible, especially in the case of axis values and captions. This figure must be improved.  

·         The conclusion and discussion are very good with a nice introduction to the application and future steps.

In conclusion: The paper is clear. The biggest advantage is the presentation of tests in real-life applications. The results presentation and graphical output are very nice. The paper needs however some editing and especially improvement in the introduction section.

Due to some small flaws, the reviewer is marking the paper for minor revisions. Hope the authors will use some suggestions to improve this otherwise very interesting and good, in case of results presentation, paper.  

Author Response

Thank you for your valuable comments. We revised the manuscript according to your suggestions. Please see the attachment.

Reviewer 2 Report

This research utilized a combination of terrestrial laser scanning (TLS) and aerial photogrammetric methods to compare point clouds and track the behavior of buildings. The advantages of non-destructive techniques like TLS and unmanned aerial systems (UAS) were also examined. By studying a building in a university campus as a case, the proposed methods successfully determined the deformations of facades over a long period of time.

1.      Abstract should be written by a coherent and fluent language due to some essential points such as research purposes, research methods, research contents and research effects, otherwise the innovation and necessity of the manuscript will not be reflected effectively. The general motivation of the work must be clearly explained in the introduction and abstract. The contribution should be clearly presented.

2.      Minor grammar and syntax issues need correction to improve readability. As an example, the sentence in line 215 of page 7 is incomprehensible. Please proofread the manuscript to avoid typos and grammatical mistakes.

3.      In the introduction section, the literature on using unmanned aerial systems for 3D modeling and monitoring is not sufficiently reviewed. It is recommended to select the following representative papers for a more detailed review:

·         Review of robot-based automated measurement of vibration for civil engineering structures. Measurement, 207 (Feb. 2023)

·         Autonomous 3D vision-based bolt loosening assessment using micro aerial vehicles. Computer-Aided Civil and Infrastructure Engineering, 00, 1– 12.

·         Vision-Based Navigation Techniques for Unmanned Aerial Vehicles: Review and Challenges. Drones 2023, 7, 89.

4.      The font sizes are not consistent in different figures. Please make sure that the font is easily readable and the same size as the main font used for the texts. For instance, the font size in Figure 2 is too big, whereas the font size in Figures 4 and 22 are too small and unreadable.

5.      The authors failed to present detailed information about the “Field Operations” in section 2.3.1. Please provide more detailed information in this section, for instance uncertainties during the flight of the UAS, type of deployed sensors on the UAS, etc.

Minor editing of English language required

Author Response

(The authors gave the same response as above.)

Reviewer 3 Report

(1) Figure.1 should be redrawn as it's current version provides little information,

(2) The authors did not state their contribution properly in the end of the introduction.

(3) Several newly developed mapping systems should be reviewed and analyzed. DOI: 10.1109/TGRS.2023.3275307

I also recommend the authors detailed the UAV operation details, like mission planning, flying height, time, overlaps, and etc.

  •  

Author Response

(The authors gave the same response as above.)

Round 2

Reviewer 2 Report

The manuscript has been carefully revised based on the reviewer's comments, and the majority comments have been addressed properly. In my opinion, the paper can be accepted in the form after the above issues are addressed.

Minor editing of English language required

Reviewer 3 Report

The authors have modified the manuscript according to my comments. It could be accepted in current form.